

Ground based characterization of spectral optical properties of haze and
Asian dust episodes under Asian continental outflow during winter 2014
Jinsang Jung[a,*], JeongAh Yu[b], Youngsook Lyu[b], Minhee Lee[b], Taekyung Hwang[b], Sangil
Lee[a]
[a]Center for Gas Analysis, Korea Research Institute of Standards and Science (KRISS),
Daejeon 34113, Republic of Korea
[b]Department of Climate and Air Quality Research, National Institute of Environmental
Research, Daejeon 34944, Republic of Korea
Running title: Optical properties in Daejeon
Last modified: November 23, 2016
Will be submitted to *Atmospheric Chemistry and Physics*
*Corresponding author: Jinsang Jung (jsjung@kriss.re.kr)





**Abstract**
An intensive field campaign was conducted in a downwind area of the Asian
continental outflow (Daejeon, Korea) during winter 2014 to characterize the spectral
optical properties of severe haze and Asian dust episodes. High concentrations of $PM_{10}$
(particulate matter with a diameter $\leq$ 10 µm) and light scattering coefficients at 550 nm,
$\sigma_{s,550}$, were observed during a long-range transport (LRT) haze episode ($PM_{10}$ = 163.9 $\pm$
25.0 µg/m$^3$; $\sigma_{s,550}$ = 503.4 $\pm$ 60.5 Mm$^{-1}$) and Asian dust episode ($PM_{10}$ = 211.3 $\pm$ 57.5
µg/m$^3$; $\sigma_{s,550}$ = 560.9 $\pm$ 151 Mm$^{-1}$). During the LRT haze episode, no significant change
in the relative contribution of $PM_{2.5}$ (particulate matter with a diameter $\leq$ 2.5 µm)
chemical components was observed as particles accumulated under stagnant
atmospheric conditions (January 13–17, 2014), suggesting that the increase in $PM_{2.5}$
mass concentration was caused mainly by the accumulation of LRT pollutants. On the
other hand, a gradual decrease in Ångström exponent ($\mathring{A}$), gradual increase in single
scattering albedo ($\omega$) and mass scattering efficiency ($MSE$) were observed during the
stagnant period, possibly due to an increase in particle size. During the Asian dust
episode, a low $PM_{2.5}/PM_{10}$ ratio and $\mathring{A}$(450/700) were observed with average values of
0.59 $\pm$ 0.06 and 1.08 $\pm$ 0.14, respectively, which were higher than those during the LRT
haze episode (0.75 $\pm$ 0.06 and 1.39 $\pm$ 0.05, respectively), indicating that $PM_{2.5}/PM_{10}$
mass ratios and $\mathring{A}$(450/700) can be used as tracers to distinguish aged LRT haze and
Asian dust under the Asian continental outflow.



## 1. Introduction

The optical property of aerosol particles is a very important parameter to understand the aerosol effects on radiative forcing and climate change. Spatiotemporal distributions of aerosol particles are needed to accurately calculate radiative forcing in the global climate system (Li et al., 2016). Atmospheric chemical transport models (CTM) are useful tools for estimating the spatial distributions and concentrations of aerosol particles on regional to global scales. In addition to CTMs, satellite remote sensing is widely used to characterize aerosol particles and their impact on climate change and air quality (van Donkelaar et al., 2010). However, both methods are uncertain due to lack of regional specific optical properties. Thus, to improve the accuracy of CTMs and satellite remote sensing, it is essential to validate these approaches using ground-based remote sensing techniques and surface optical measurements.

Severe haze over China can influence the air quality of downwind areas of the Asian continent and regional environments over the East Asia through long-range transport (LRT) by the prevailing westerly (Aikawa et al., 2010; Jung and Kim, 2011; Kaneyasu et al., 2014; Jung et al., 2015). LRT haze can also affect the regional radiation budget directly by scattering or absorbing solar radiation and indirectly by altering the physical properties of clouds and the efficiency of precipitation (Ramanathan et al., 2007; Gao et al., 2014; Jeong et al., 2014; Jung et al., 2015). Zhang et al. (2007) reported that the Asian pollution outflow influences precipitation over the North Pacific. To investigate the impact of LRT haze on regional environments over downwind areas of the Asian continental outflows, it is necessary to characterize the optical properties of LRT haze.

The single scattering albedo ($\omega$) is the key parameter used to determine the aerosol effect on radiative forcing and climate change (IPCC, 2013). Thus, accurate



measurements of scattering and absorption properties of aerosol particles are important
for the better estimation of aerosol radiative forcing. Spectral $\omega$ and the backscattering
ratio, defined as the ratio of light scattered in the backward hemisphere to the total light
scattered, also provide information for the accurate determination of aerosol radiative
forcing (Gopal et al., 2014). However, *in situ* observations of spectral aerosol optical
properties under Asian continental outflows are rare; thus, an intensive characterization
of aerosol optical properties is needed.
In Shanghai, China, the $\omega$ measured at the surface shows a weak seasonal variation
whereas a ground-based remote sensing technique shows the highest $\omega$ during the fall
season. The $\omega$ measured by ground-based remote sensing (0.9−0.93) is ~10 % higher
than values measured at the surface (0.8−0.9) (Cheng et al., 2015). From one year's
worth of observations in Seoul, Korea, a trend of increasing $\omega$ with wavelength was
observed during Asian dust events whereas little spectral dependence of $\omega$ was observed
during LRT haze events (Jung et al., 2010). During the Campaign of Air Quality
Research in Beijing 2006 (CAREBeijing-2006), $\omega$ was found to be closely related to the
inflow of air to Beijing. Relatively low $\omega$ (<0.8) was observed for the air mass
originated from the north of Beijing whereas relatively high $\omega$ was observed for the air
mass originated from the south of Beijing (Garland et al., 2009).
The objective of this study is to characterize the spectral optical properties of the
LRT haze and Asian dust originating from the Asian continent during winter 2014.
Integrated chemical and optical measurements of aerosol particles were carried out at
Daejeon, Korea during January 2014 to characterize the optical properties of different
types of haze. Temporal variations in spectral optical properties under stagnant





atmospheric conditions are discussed with reference to aerosol chemical composition.
From identified Asian continental outflows, we also investigated the wavelength
dependence of aerosol optical properties.

2.  Experimental Methods
2.1 Measurement site

Online measurements of aerosol optical properties and daily $PM_{2.5}$ (particulate

matter with a diameter ≤ 2.5 μm) sampling were conducted at an air quality monitoring
station in the mega-city of Daejeon, central Korea (36.19°N, 127.24°E) during January
8−31, 2014 (Fig. 1). Because Daejeon is located downwind of Asian continental
outflows, it is frequently affected by long-range transported pollutants and Asian dust
(Jung et al., 2016). Light scattering and absorption coefficients were continuously
measured inside a monitoring building (~15 m above the ground) of the National
Institute of Environmental Research in Korea. $PM_{2.5}$ samples were collected on pre-
baked quartz fiber filters (Pall-Life Sciences, 47 mm diameter) at a flow rate of 16.7 L
$min^{-1}$. An aerosol sampler (APM Korea, model PMS-103) was installed on the rooftop
of the monitoring building. Before and after sampling, filter samples were stored in a
freezer at −20 ℃ wrapped with aluminum foil. A total of 23 filter samples were
collected. Additionally, field blank filters were collected before and after the sampling
period. Hourly precipitation data were obtained from a nearby weather monitoring
station of the Korea Meteorological Agency.

2.2 Online measurement of aerosol chemical composition

$PM_{10}$ and $PM_{2.5}$ mass concentrations were measured by a beta-attenuation monitor



(Met One Instruments, BAM 1020) with an hourly interval. The detection limit of the
beta-attenuation technique is reported as 3.6 µg m$^{-3}$ by the manufacturer. Hourly PM$_{10}$
potassium (K) concentrations were continuously measured by X-ray fluorescence (XRF)
(Cooper Environmental Service (CES), Model Xact 620). The air samples were
introduced through a PM$_{10}$ inlet at a flow rate of 16.7 L min$^{-1}$ and drawn through filter
tape. The online Xact 620 monitor was calibrated using thin film standards for each
element of interest, which was provided by CES. These standards were manufactured by
depositing vapor-phase elements on blank Nuclepore (Micromatter Co.). For a 1 hr time
resolution, the minimum detection limit for K has been reported to be 0.81 ng m$^{-3}$ (Park
et al., 2014).
Online measurements of PM$_{2.5}$ organic carbon (OC) and elemental carbon (EC)
were conducted using a semi-continuous carbon analyzer (Sunset Laboratory Inc.,
Model RT3140) based on the thermal-optical transmittance (TOT) protocol for
pyrolysis correction and the NIOSH (National Institute for Occupational Safety and
Health) 5040 method temperature profile (Birch and Cary, 1996; Jung et al., 2010).
Measurement condition of the carbon analyzer was described in detail by Jung et al.
(2016). The detection limit of both OC and EC was 0.5 µg C m$^{-3}$ for 1 hr time
resolution, as reported by the manufacturer. The uncertainty of OC and EC
measurements has been reported to be 5% (Polidori et al., 2006).

2.3 Online measurement of aerosol optical properties
Light scattering coefficients ($\sigma_s$) and hemispheric backscattering coefficients ($\sigma_{bs}$)
of aerosol particles at three wavelengths ($\lambda$ = 450, 550, and 700 nm) were continuously
measured using an integrating nephelometer (TSI inc., model 3563). The nephelometer



was operated at a flow rate of 5 L min$^{-1}$ with a 5-min averaging time. The clean air and
span gas (pure $CO_2$) calibrations were carried out every hour and once a month,
respectively. The uncertainty of the nephelometer measurements was determined to be
less than 2% with a 5-min interval. For a 5-min resolution, the detection limits of $\sigma_s$
were determined to be 6, 3, and 3 Mm$^{-1}$ at 450, 550, and 700 nm, respectively,
calculated as $3\sigma$ of the clean air measurement. Systematic biases caused by angular
truncation errors and a non-Lambertian light source were corrected for scattering
measurement data using the the Ångström exponents of $\sigma_s$ (Anderson et al., 1996;
Anderson and Ogren, 1998; Garland et al., 2009). The corrected systematic biases were
~12% of the measured values. The relative humidity (RH) of the sampled air inside the
nephelometer chamber was 21 ± 10%.
The optical attenuation coefficients ($\sigma_{ATN}$) of aerosol particles were measured using
the Aethalometer (Magee Scientific, Model AE31) at seven wavelengths (370, 470, 520,
590, 660, 880, and 950 nm) (Hansen, 2005). Air samples were drawn through the PM$_{2.5}$
cyclone (BGI Inc., SCC1.829) at a flow rate of 4 L min$^{-1}$. The light absorption
coefficient ($\sigma_a$) was retrieved from $\sigma_{ATN}$ as described by Jung et al. (2010), by
considering the "shadowing effect" and multiple scattering within the filter. The
detection limit of the aethalometer $\sigma_a$, defined as $3\sigma$ of the dynamic blank, was
determined to be 2 Mm$^{-1}$. The measurement uncertainty of the Aethalometer is reported
to be ±5% by the manufacturer (Hansen, 2005).

2.4  Chemical analysis of filter samples
A quarter of each filter sample was extracted with 10 mL of ultrapure water under




ultrasonication (for 30 min). Water extracts were then passed through a disk filter
(Millipore, Millex-GV, 0.45 mm) to remove filter debris and water-insoluble particles.
Water extracts were stored in a refrigerator at 4 ℃ prior to analysis. The total organic
carbon (TOC) level of the ultrapure water was maintained below 4 ppb using a Labpure
S1 filter and an ultraviolet (UV) lamp (ELGA, PureLab Ultra).
Water-soluble inorganic ions were analyzed using an ion chromatograph (Thermo
Fisher Scientific, Dionex ICS-15000). Analytical conditions of anions ($Cl^-$, $NO_3^-$, $SO_4^{2-}$)
and cations ($Na^+$, $NH_4^+$, $K^+$, $Ca^{2+}$, $Mg^{2+}$) were described in detail by Jung et al. (2016).
The detection limits of $Cl^-$, $NO_3^-$, and $SO_4^{2-}$, which are defined as 3 times the standard
deviation of field blanks, were determined to be 0.02, 0.01 and 0.11 $\mu g\ m^{-3}$, respectively.
The analytical error of $Cl^-$, $NO_3^-$ and $SO_4^{2-}$ measurements was 2.0%, 1.7%, and 2.3%,
respectively. The detection limits of $NH_4^+$ and $K^+$ were determined to be 0.03 and 0.006
$\mu g\ m^{-3}$, respectively. The analytical errors of $NH_4^+$ and $K^+$ were determined to be 1.4%
and 0.73%, respectively.

2.5 Satellite RGB images and air mass backward trajectories
Moderate Resolution Imaging Spectro-radiometer (MODIS) satellite images were
obtained from the NASA/MODIS web site (https://modis.gsfc.nasa.gov/). Air mass
backward trajectories ending at the measurement site were calculated for heights of 200,
500 and 1000 m above ground level (AGL) using the HYSPLIT (HYbrid Single-
Particle Lagrangian Trajectory) model (Draxler and Rolph, 2016; Rolph, 2016). All
back trajectories were ended at 00:00 UTC and 12:00 UTC (09:00 LT and 21:00 LT,
respectively) and extended 96 hr backwards.






### 2.6 Intensive optical properties

#### 2.6.1    Ångström exponent of aerosol light scattering

The wavelength dependent aerosol scattering can be expressed by a power law (Ångström, 1929) as follows:

$$\sigma_{s,\lambda} = \sigma_{s,\lambda r}\left(\frac{\lambda}{\lambda_r}\right)^{-\text{Å}} \tag{1}$$

where $\sigma_{s,\lambda r}$ is the scattering coefficient at a reference wavelength $\lambda_r$ and $\text{Å}$ is the Ångström exponent. The Ångström exponent can be retrieved from the slope of a double-logarithmic plot of $\sigma_s$ versus $\lambda$ as follows:

$$\text{Å}\left(\frac{\lambda_1}{\lambda_2}\right) = -\frac{\log\left(\frac{\sigma_{s,\lambda 1}}{\sigma_{s,\lambda 2}}\right)}{\log\left(\frac{\lambda_1}{\lambda_2}\right)} \tag{2}$$

#### 2.6.2    Backscattering fraction, single scattering albedo, and mass scattering efficiency

The backscattering coefficient is defined as the scattered light intensity in the backward hemisphere of the particle (90°–180°) (Anderson and Ogren, 1998). The backscattering ratio is used to derive the slope of the particle size distribution and also provides an estimate of the bulk refractive index of particles in the atmosphere (Gopal et al., 2014). The hemispheric backscattering fraction, $b_\lambda$, is defined as the ratio of the backscattering coefficient to the total scattering coefficient at a given wavelength ($\lambda = $ 450, 550, and 700 nm), calculated as

$$b_\lambda = \frac{\sigma_{bs,\lambda}}{\sigma_{s,\lambda}} \tag{3}$$

The single scattering albedo, $\omega_\lambda$, is the ratio of the scattering coefficient to the extinction coefficient at a given wavelength. Here, $\omega_\lambda$ at a certain $\lambda$ can be calculated as





follows:

$$\omega_\lambda = \frac{\sigma_{s,\lambda}}{\sigma_{s,\lambda}+\sigma_{a,\lambda}}$$     (4)

Because $\sigma_a$ was not measured at 550 nm by an aethalometer, $\sigma_a$ at $\lambda$ = 520 nm is
converted to $\sigma_a$ at $\lambda$ = 550 nm as follows:

$$\sigma_{a,500} = \sigma_{a,520} * \left(\frac{\lambda(550\,nm)}{\lambda(520\,nm)}\right)^{-\alpha}$$     (5)

where $\alpha$ is the absorbing Ångström exponent, which was determined from spectral
aerosol light absorption as follows:

$$\alpha = -\frac{\log(\sigma_{a,590})-\log(\sigma_{a,520})}{\log(590\,nm)-\log(520\,nm)}$$     (6)

The mass scattering efficiency, $MSE_\lambda$, is the ratio of the scattering coefficient to the
mass concentrations at a given wavelength, expressed as

$$MSE_\lambda = \frac{\sigma_{s,\lambda}}{PM_{2.5}\,mass}$$     (7)


3.  Results and Discussion
3.1   Temporal variations in PM mass and light scattering coefficient ($\sigma_s$)
Figure 2 shows temporal variations in wind speed and hourly precipitation, $PM_{10}$ and
$PM_{2.5}$ mass, the $PM_{2.5}/PM_{10}$ mass ratio, and the light scattering coefficient ($\sigma_s$) at the
measurement site in Daejeon during January 8–31, 2014. The $PM_{10}$ mass concentrations
ranged from 19 to 270 μg m$^{-3}$ with an average of 83 ± 42 μg m$^{-3}$, and $PM_{2.5}$ mass
concentrations ranged from 8 to 147 μg m$^{-3}$ with an average of 57 ± 30 μg m$^{-3}$ during
the measurement period. The average $PM_{2.5}$ mass concentration in this study is much
higher than the US EPA NAAQS (National Ambient Air Quality Standards) for $PM_{2.5}$ of
35 μg m$^{-3}$ (24 hr average). Average $PM_{2.5}/PM_{10}$ mass ratios ranged from 0.41 to 0.93



with an average of 0.68 ± 0.1. $\sigma_s$ at 550 nm ranged from 12.7 to 678.4 Mm$^{-1}$ with an
average of 189.1 ± 142.0 Mm$^{-1}$. The average $\sigma_s$ in this study is comparable with the
annual mean of 217 Mm$^{-1}$ measured in the Shanghai region, China during 2010–2012
(Cheng et al., 2015) but is lower than the annual mean of 360 Mm$^{-1}$ in the Beijing
region, China measured during 2009–2010 (Jing et al., 2015). Because light scattering is
caused mainly by aerosol particles and the scattering measurements of the present study
were performed under dry conditions (RH < 30%), similar temporal patterns were
observed for PM mass and $\sigma_s$ (Fig. 2).
As shown in Fig. 2, three haze episodes were observed on 12, 17, and 20 January
2014 with peak PM$_{10}$ mass concentrations of 173, 210, and 270 μg m$^{-3}$, respectively.
PM$_{2.5}$/PM$_{10}$ mass ratios during the episodes were measured as 0.71, 0.69, and 0.54,
respectively, during the three episodes. The first and second haze episodes were caused
mainly by the accumulation of pollutants for 3–4 days under stagnant atmospheric
conditions with relatively low wind speed (Fig. 2). After 3–4 days of aerosol
accumulation, PM mass concentrations showed a sharp decrease with relatively high
wind speeds (> 2 m/sec). A sharp increase in PM$_{10}$ mass was observed during the third
episode when a relatively high wind speed was observed (Fig. 2a). A similar temporal
pattern was observed for $\sigma_s$ and PM$_{10}$ mass concentrations during the three haze
episodes. The light scattering coefficient at 550 nm reached peak values of 494.2, 594.4,
and 678.4 Mm$^{-1}$ during the first, second, and third episodes, respectively (Fig. 2d).
During the first and second haze episodes, no precipitation was observed whereas
before and after the third haze episode light precipitation was observed with an hourly
average of 0.5−1.5 mm/hr. Sharp decrease of $\sigma_s$ and PM$_{10}$ mass concentrations during
the third haze episode was mainly attributed to precipitation. However, the first and





second haze episodes were not influenced by precipitation.

3.2   Single scattering albedo ($\omega$), Ångström exponent ($\mathring{A}$), and backscattering fraction

($b$)

Figure 3 shows temporal variations in aerosol optical properties, including $\sigma_s$, $\mathring{A}$,

backscattering fraction ($b$), and $\omega$. The $\mathring{A}$ value between 450 and 700 nm ($\mathring{A}(450/700)$)
ranged from 0.94 to 1.99 with an average of 1.60 ± 0.19, which is comparable to the
$\mathring{A}(450/550)$ value of 1.59 ± 0.21 and $\mathring{A}(550/700)$ value of 1.61 ± 0.19 listed in Table 1.
The $\mathring{A}(450/700)$ value obtained in this study is slightly higher than that obtained in
Beijing, China during summer 2006 (1.42 ± 0.19; Garland et al., 2009) and that
obtained in Guangzhou, China during summer 2006 (1.51 ± 0.20; Garland et al., 2008).
Because $\mathring{A}$ is negatively correlated with particle diameter (Eck et al., 1999), the slightly
higher $\mathring{A}$ observed in this study compared with those from Mainland China implies
larger aerosol particles in this study.

During the measurement period, $b$ at 550 nm ($b_{550}$) ranged from 0.08 to 0.17 with an

average of 0.12 ± 0.02, which is comparable with $b_{450}$ (0.12 ± 0.02) but slightly lower
than $b_{700}$ (0.15 ± 0.02). Similar patterns of $b$ with wavelength were observed in Beijing
owing to a decrease in particle size with increasing wavelength (Garland et al., 2009).
The $\omega$ at 550 nm ($\omega_{550}$) ranged from 0.58 to 0.95 with an average of 0.85 ± 0.07, which
is comparable with $\omega_{450}$ (0.85 ± 0.07) and $\omega_{700}$ (0.84 ± 0.08). The average $\omega_{550}$ is close
to the values reported from other locations in and around Beijing and Guangzhou ($\omega_{550}$
= 0.82–0.85) (Bergin et al., 2001; Andreae et al., 2008; Cheng et al., 2008; Garland et
al., 2008, 2009).

Dynamic temporal patterns in $\mathring{A}$, $b$, and $\omega$ were observed during the measurement





period (Fig. 3). Gradual decreases in $Å$ with increasing $\sigma_s$ were observed during the first
and second haze episodes, whereas a sharp decrease in $Å$ was observed with increasing
$\sigma_s$ during the third episode. It was also found that $b$ was negatively correlated with $\sigma_s$
during the three episodes. Meanwhile, $\omega$ increased gradually with $\sigma_s$ during the first and
second episodes. These results indicate that temporal variations in $Å$, $b$, and $\omega$ are
closely related to those in $\sigma_s$. In this study, $Å$ and $b$ were negatively correlated with $\sigma_s$
whereas $\omega$ was positively correlated with $\sigma_s$.
Figure 4 clearly shows that $\omega_{550}$ increases with $\sigma_{s,550}$. When $\omega_{550}$ was less than 200
$Mm^{-1}$, $\omega_{550}$ varied widely from 0.6 to >0.9. The $Å(450/700)$ value increased with $\sigma_{s,550}$
when $\sigma_{s,550}$ was lower than ~150 $Mm^{-1}$. However, when $\sigma_{s,550}$ was higher than ~150
$Mm^{-1}$, $Å(450/700)$ gradually decreased with increasing $\sigma_{s,550}$. Figure 5a shows a scatter
plot of $\omega_{550}$ versus $b_{550}$ as a function of $\sigma_{s,550}$, where $\omega_{550}$ is observed to decrease as $b_{550}$
increases. A scatter plot of $b_{550}$ versus $Å(450/700m)$ is shown in Fig. 5b. A positive
correlation is observed between $Å(450/700)$ and $b_{550}$ when $\sigma_{s,550}$ is higher than 200 $Mm^{-}$
$^1$, whereas a poor correlation is observed when $\sigma_{s,550}$ is lower than 200 $Mm^{-1}$. In
addition, a relatively small $b_{550}$ is observed as $\sigma_{s,550}$ increases (Fig. 5a and b).

3.3   Aerosol optical properties during severe haze episodes
3.3.1 Classification of haze episodes
As shown in Fig. 2b, three haze episodes were observed during 11–12, 14–17, and 20
January 2014. This study focused on the second and third haze episodes, which peaked
on 17 and 20 January 2014. Figure 6 shows MODIS RGB images during 14–17 January
2014. A dense haze layer is clearly seen over East China during 14 January. This layer
moved slowly to the Korean Peninsula from 15 to 17 January. Air mass backward



trajectories ending at the measurement site also show the transport of air masses from
East China to the Korean Peninsula on 17 January 2014, as shown in Fig. 7a. During the
second haze episode, very low wind speeds of <1 m sec$^{-1}$ were observed (Fig. 2a). Thus,
the second haze episode is classified as a period of accumulation of LRT pollutants from
the Asian continent (LRT haze).
During the third haze episode on 20 January, very high concentrations of K
(maximum: 4.9 μg m$^{-3}$, average: 2.2 ± 2.3 μg m$^{-3}$) were observed (Table 2). The air
mass backward trajectory for 20 January clearly shows that the air mass originating
from the Nei Mongol desert area had an impact on the Korean Peninsula (Fig. 7b).
During the third haze episode, relatively high wind speeds of >2 m sec$^{-1}$ were observed
(Fig. 2a). Thus, the third haze episode is classified as an Asian dust episode.

3.3.2 Temporal variations in the chemical and optical properties of LRT haze
Figure 8 shows temporal variations in the chemical composition of PM$_{2.5}$ during the
LRT haze episode (14–17 January 2014). As mentioned above, the LRT haze episode
was caused mainly by the accumulation of long-range transported pollutants from the
Asian continent. Gradual increases in total PM$_{2.5}$ mass were observed during the LRT
haze episode (Fig. 8a). The relative contribution of PM$_{2.5}$ chemical composition is also
shown in Fig. 8b. Organic aerosol (OA) dominated the PM$_{2.5}$ mass composition,
followed by NO$_3^-$, SO$_4^{2-}$ and NH$_4^+$. Even though a small decrease in OA mass fraction
was observed during 15 January, the mass fractions of the major PM$_{2.5}$ chemical
components were invariant from 14 to 17 January. These results suggest that the
increase in PM$_{2.5}$ mass concentration observed during the LRT haze episode was
caused mainly by the accumulation of LRT pollutants.





Figure 9 shows temporal variations in the daily average intensive optical properties of the LRT haze. The $Å(450/700)$ and $b_{550}$ values decreased during the accumulation period from 14 to 17 January while $MSE_{550}$ and $\omega_{550}$ increased. Average $Å(450/700)$ decreased from $1.74 \pm 0.09$ on 14 January to $1.39 \pm 0.05$ on 17 January. Average $b_{550}$ decreased from $0.15 \pm 0.01$ on 14 January to $0.10 \pm 0.003$ on 17 January. Average $MSE_{550}$ of $PM_{10}$ increased from $1.73 \pm 0.40$ $m^2$ $g^{-1}$ on 14 January to $3.11 \pm 0.46$ $m^2$ $g^{-1}$ on 17 January. An increase in $MSE_{550}$ with increasing PM mass concentration during the haze episodes was also observed in Beijing and Guangzhou, China during summer 2006 (Jung et al., 2009a, b). For example, in Beijing the $MSE_{550}$ of $PM_{10}$ increased from $1.4 \pm 0.89$ $m^2$ $g^{-1}$ during relatively clean conditions to $3.1 \pm 0.9$ $m^2$ $g^{-1}$ during relatively polluted conditions (Jung et al., 2009a). At most monitoring sites in the United States, dry $MSE$ increased with increasing mass concentration (IMPROVE, 2006).

Average $\omega_{550}$ increased from $0.81 \pm 0.07$ on 14 January to $0.90 \pm 0.03$ on 17 January. A similar pattern was observed as pollution increased in Beijing during summer 2006 (Jung et al., 2009a). Average $\omega_{550}$ increased from ~0.75 during relatively clean conditions to ~0.86 during relatively polluted conditions in Beijing during summer 2006 owing to an increase in $SO_4^{2-}$, $NO_3^-$, $NH_4^+$, and organic aerosols (Jung et al., 2009a). Because EC is a strong light-absorbing aerosol, changes in EC mass fraction in $PM_{2.5}$ mass can be used as an indicator of $\omega$. As shown in Fig. 8b, EC mass fraction in $PM_{2.5}$ was invariant from 14 to 17 January. These results indicate that an increase in mass concentration of secondary aerosols such as $SO_4^{2-}$, $NO_3^-$, $NH_4^+$, and secondary organic aerosol cannot explain the increase in $\omega_{550}$ under stagnant conditions during the LRT haze episode. On the other hand, an increase in $MSE_{550}$ under stagnant



conditions (Fig. 9b) can enhance $\omega_{550}$, resulting in an increase in $\omega_{550}$ under stagnant
conditions.
The amount of light scattered by aerosol particles can be accurately estimated using
Mie theory when the size distribution and refractive index of the particles are known
(Mie, 1908; Hess et al., 1998; Seinfeld and Pandis, 1998). Light scattering efficiencies
of $(NH_4)_2SO_4$ and organic aerosols at 550 nm were calculated using Mie theory using
refractive indices for $1.53–0i$ and $1.55–0i$, respectively (Liu et al., 2009), as shown in
Fig. 10. Light scattering efficiencies of $(NH_4)_2SO_4$ and organic aerosols at 550 nm
increase as particle size increases to 600 nm.
Freshly formed aerosol particles have a diameter ($D_p$) of less than 100 nm (Yue et al.,
2010) and grow into the accumulation mode (100 nm $< D_p <$ 1000 nm) through the
condensation of gas vapors or coagulation (collisions between particles; Seinfeld and
Pandis, 1998). Thus, larger particles (in the accumulation mode) are observed under
polluted stagnant conditions. An increase in $D_p$ under stagnant conditions can enhance
light scattering, resulting in an increase in *MSE*. *Å* and *b* are also closely related to the
size of aerosol particles. For example, Eck et al. (1999) reported that coarse mode
particles had relatively low *Å* compared with fine mode particles. Nemesure et al.
(1995) reported that the forward scattering fraction increases as particle size increases,
resulting in a decrease in *b*. This suggests that the temporal variations in intensive
optical properties shown in Fig. 9 are closely related to the change in size of aerosol
particles under stagnant conditions.

3.3.3 Inter-comparison of the aerosol optical properties of LTP haze versus Asian dust

particles



Optical properties of the LRT haze and Asian dust are compared in Fig. 11 and
summarized in Table 2. For this comparison, data obtained on 17 January were used to
represent aged LRT haze. Elevated K concentrations were observed during the Asian
dust episode, with an average of $2.2 \pm 2.3$ μg m$^{-3}$. Similar levels of $PM_{2.5}$ mass were
obtained during the LRT haze and Asian dust episodes, whereas much higher $PM_{10}$
mass was obtained during the Asian dust episode compared with the LRT haze episode
(Table 2), resulting in higher $PM_{2.5}/PM_{10}$ mass ratios during the LRT haze episode (0.75
$\pm$ 0.0) compared with the Asian dust episode ($0.59 \pm 0.06$). Higher $EC/PM_{10}$ mass ratios
were observed during the LRT haze episode with an average of $0.033 \pm 0.00$ compared
with the Asian dust episode ($0.026 \pm 0.003$). $PM_{2.5}/PM_{10}$ mass ratios and $EC/PM_{10}$ mass
ratios during the Asian dust episode were higher than those obtained in Seoul, Korea
during severe Asian dust episodes in 2007–2008 ($PM_{2.5}/PM_{10} < 0.4$; $EC/PM_{10} < 0.013$).
In addition, high $PM_{2.5}$ mass concentrations during the Asian dust episode in this study
suggest that Asian dust particles mixed with LRT haze originating from anthropogenic
emissions had an impact on the measurement site on 20 January.
Similar levels of $\sigma_s$ were observed during the LRT haze ($503.4 \pm 60.5$ Mm$^{-1}$) and
Asian dust episode ($560.9 \pm 151$ Mm$^{-1}$) (Fig. 11a). The $\omega_{550}$ values obtained for the two
episodes were comparable, with averages of $0.91 \pm 0.03$ and $0.92 \pm 0.0$ observed during
the LRT haze and Asian dust episodes, respectively. However, a higher light absorption
coefficient ($\sigma_{a,550}$) was obtained during the LRT haze episode ($51.9 \pm 21.9$ Mm$^{-1}$)
compared with the Asian dust episode ($39.4 \pm 7.3$ Mm$^{-1}$). Higher $\mathring{A}$(450/700) was
obtained during the LRT haze episode (average of $1.39 \pm 0.05$) compared with the
Asian dust episode ($1.08 \pm 0.14$), due mainly to the relatively large size distribution
during the Asian dust episode. The results of this study suggest that $PM_{2.5}/PM_{10}$ mass



ratios and $\mathring{A}(450/700)$ can be used as tracers to distinguish aged LRT haze and Asian
dust based on differences in the particle size distribution.

4. Conclusion
An intensive field campaign was conducted at an area downwind of the Asian
continental outflow (Daejeon, Korea) during winter 2014 to characterize the spectral
optical properties of severe haze episodes. Dynamic temporal patterns of aerosol optical
properties were observed during the measurement period. During the stagnant period
(January 13–17, 2014), after long-range transport of haze from the Asian continent, no
significant change in the mass fraction of $PM_{2.5}$ chemical composition was observed,
with the highest fraction being organic aerosol, followed by $NO_3^-$, $NH_4^+$, and $SO_4^{2-}$. On
the other hand, a gradual decrease in Ångström exponent ($\mathring{A}$) and gradual increases in
single scattering albedo ($\omega$) and mass scattering efficiency ($MSE$) were observed during
the stagnant period. Mie calculations suggest that the increase in aerosol particle
diameter under stagnant conditions enhanced light scattering, resulting in an increase in
$MSE$. It is also suggested that the increase in $MSE$ under stagnant conditions enhanced
$\omega$. These results imply that changes in particle size rather than chemical composition
during the stagnant period is the dominant factor affecting the aerosol optical properties.
During the Asian dust episode, very high values of $PM_{10}$ mass and light scattering
coefficient at 550 nm, $\sigma_{s,550}$, were observed with averages of 211.3 ± 57.5 µg m$^{-3}$ and
560.9 ± 151 Mm$^{-1}$, respectively. The $\omega_{550}$ during the LTP haze and Asian dust episodes
were comparable, with averages of 0.91 ± 0.03 and 0.92 ± 0.0, respectively, implying
that aged LRT pollutants and Asian dust particles have similar $\omega$. A relatively small
$PM_{2.5}/PM_{10}$ ratio and $\mathring{A}(450/700)$ were observed during the Asian dust episode



compared with those during the LRT haze episode, indicating that $PM_{2.5}/PM_{10}$ mass
ratios and $\mathring{A}(450/700)$ can be used as tracers to distinguish aged LRT haze and Asian
dust.

Acknowledgements
This work was conducted by a co-research project of the National Institute of
Environmental Research (NIER) and the Korean Research Institute of Standards and
Science (KRISS). This study was funded by the Korean Meteorological Administration
Research and Development Program under grant KMIPA 2015-5020.




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





Table 1. Summary of aerosol optical parameters observed in Daejeon, Korea during
January 2014.

| Component | Unit | Min–Max (Average ± S.D.) |
|---|---|---|
| Light scattering coefficient, $\sigma_{s,450}$ | Mm$^{-1}$ | 16.5–805.0 (256.9 ± 183.7) |
| $\sigma_{s,550}$ | Mm$^{-1}$ | 12.7–678.4 (189.1 ± 142.0) |
| $\sigma_{s,700}$ | Mm$^{-1}$ | 9.3–531.6 (129.1 ± 101.3) |
| Backscattering coefficient, $\sigma_{bs,450}$ | Mm$^{-1}$ | 2.4–77.2 (27.6 ± 16.9) |
| $\sigma_{bs,550}$ | Mm$^{-1}$ | 1.7–61.3 (21.3 ± 13.3) |
| $\sigma_{bs,700}$ | Mm$^{-1}$ | 1.4–57.2 (17.8 ± 11.7) |
| Å ngström Exponent of $\sigma_{s}$, Å(450/550) | | 0.85–2.06 (1.59 ± 0.21) |
| Å(450/700) | | 0.94–1.99 (1.60 ± 0.19) |
| Å(550/700) | | 1.0–1.97 (1.61 ± 0.19) |
| Hemispheric backscattering fraction, $b_{450}$ | | 0.08–0.17 (0.12 ± 0.02) |
| $b_{550}$ | | 0.08–0.17 (0.12 ± 0.02) |
| $b_{700}$ | | 0.1–0.19 (0.15 ± 0.02) |
| Single scattering albedo, $\omega_{450}$ | | 0.57–0.95 (0.85 ± 0.07) |
| $\omega_{550}$ | | 0.58–0.95 (0.85 ± 0.07) |
| $\omega_{700}$ | | 0.56–0.95 (0.84 ± 0.08) |







Table 2. Comparison of PM mass, chemical components, and intensive optical
properties during long-range transported (LRT) haze and Asian dust episodes observed
at Daejeon in Korea during January 2014.

| | LRT haze[a] | Asian dust[b] |
|---|---|---|
| | Min–Max (Average ± S.D.) | |
| $PM_{10}$ ($\mu g\ m^{-3}$) | 133–210 (163.9 ± 25.0) | 126–270 (211.3 ± 57.5) |
| $PM_{2.5}$ ($\mu g\ m^{-3}$) | 100–145 (121.6 ± 12.8) | 86–147 (121.5 ± 22.7) |
| $PM_{2.5}/PM_{10}$ ratio | 0.68–0.84 (0.75 ± 0.06) | 0.48–0.68 (0.59 ± 0.06) |
| $EC/PM_{10}$ ratio | 0.026–0.047 (0.033 ± 0.006) | 0.023–0.032 (0.026 ± 0.003) |
| K ($\mu g\ m^{-3}$) | 0.1–1.5 (0.8 ± 0.5) | 0.02–4.9 (2.2 ± 2.3) |
| $\sigma_{s,550}$ ($Mm^{-1}$) | 358.8–594.4 (503.4 ± 60.5) | 276.1–678.4 (560.9 ± 151) |
| $\sigma_{a,550}$ ($Mm^{-1}$) | 29.3–105.4 (51.9 ± 21.9) | 29.4–46.1 (39.4 ± 7.3) |
| Å(450/700) | 1.30–1.47 (1.39 ± 0.05) | 0.94–1.25 (1.08 ± 0.14) |
| $\omega_{550}$ | 0.84–0.94 (0.91 ± 0.03) | 0.90–0.94 (0.92 ± 0.02) |

[a]LRT haze: 17 January 2014, 00:00–23:00 LT
[b]Asian dust: 20 January 2014, 13:00–18:00 LT




Figure captions

Fig. 1. Map of the measurement site (36.19° N, 127.24° E) in Daejeon, Korea (base map
is from Google Maps).
Fig. 2. Temporal variations in (a) wind speed and hourly precipitation, (b) $PM_{2.5}$ and
$PM_{10}$ mass concentrations, (c) $PM_{2.5}/PM_{10}$ mass ratio, and light scattering
coefficient ($\sigma_s$) at 450, 550, and 700 nm at the Daejeon site during January 2014.
Fig. 3. Temporal variations in (a) $\sigma_{s,550}$, (b) the Ångström exponent of $\sigma_s$ (Å), (c) the
backscattering fraction ($b$), and single scattering albedo ($\omega$) at 450, 550, and 700
nm. Å(450/550) represents the Ångström exponent calculated from $\sigma_s$ at 450 and
550 nm.
Fig. 4. Scatter plot of $\sigma_{s,550}$ versus (a) $\omega_{550}$ and (b) Å(450/700) during the entire
measurement period.
Fig. 5. Scatter plots of (a) $b_{550}$ versus $\omega_{550}$ and (b) Å(450/700) versus $b_{550}$ as a function
of $\sigma_{s,\,550}$.
Fig. 6. MODIS RGB images over East Asia during 14−17 January 2014.
Fig. 7. Air mass backward trajectories arriving at the measurement site on (a) 16 and (b)
20 January 2014. Red, blue, and green lines represent backward trajectories arriving
at heights of 200, 500, and 1000 m, respectively.
Fig. 8. Temporal variations in (a) mass concentrations of $PM_{2.5}$ chemical components
and (b) $PM_{2.5}$ mass fractions of major components during 14−17 January 2014.
Fig. 9. Temporal variations in (a) daily average Å(450/700) and $b_{550}$, (b) mass scattering
efficiency at 550 nm ($MSE_{550}$), and (c) $\omega_{550}$ during 14−17 January 2014.
Fig. 10. Scattering efficiency of $(NH_4)_2SO_4$ and organic aerosols as a function of



594  particle diameter, as calculated from Mie theory.

595 Fig. 11. Comparison of (a) average $\sigma_{s,550}$ during the severe long range transported haze

596  episode (17 January) and during the Asian dust episode (20 January). Comparisons

597  of $\mathring{A}(450/700)$, $b_{550}$, and $\omega_{550}$ are shown in (b), (c), and (d), respectively.







600                                                                                 Figure 1

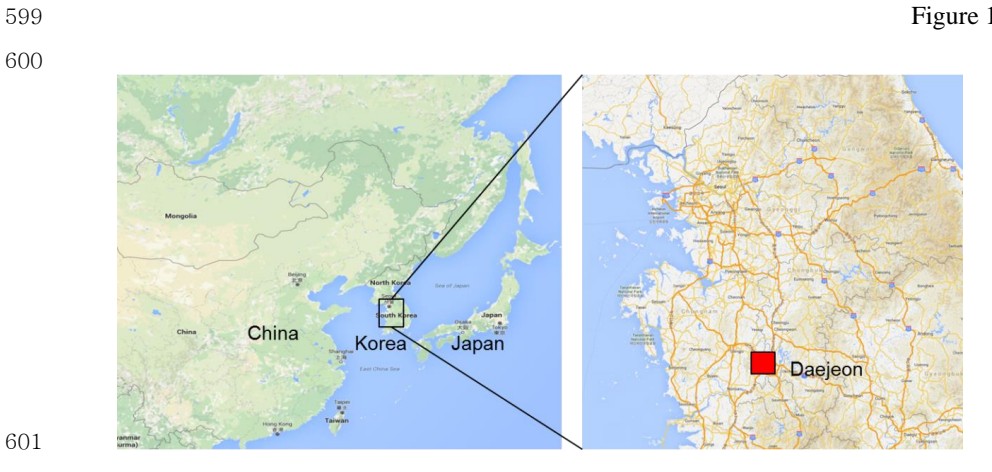







603                                                                                      Figure 2

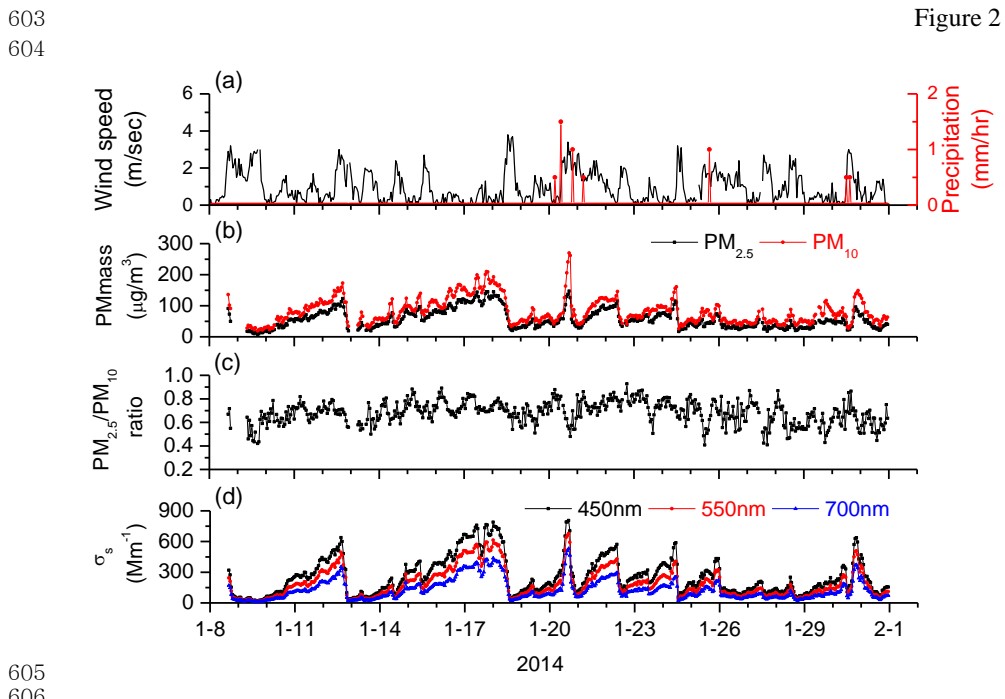




607                                                                                                    Figure 3

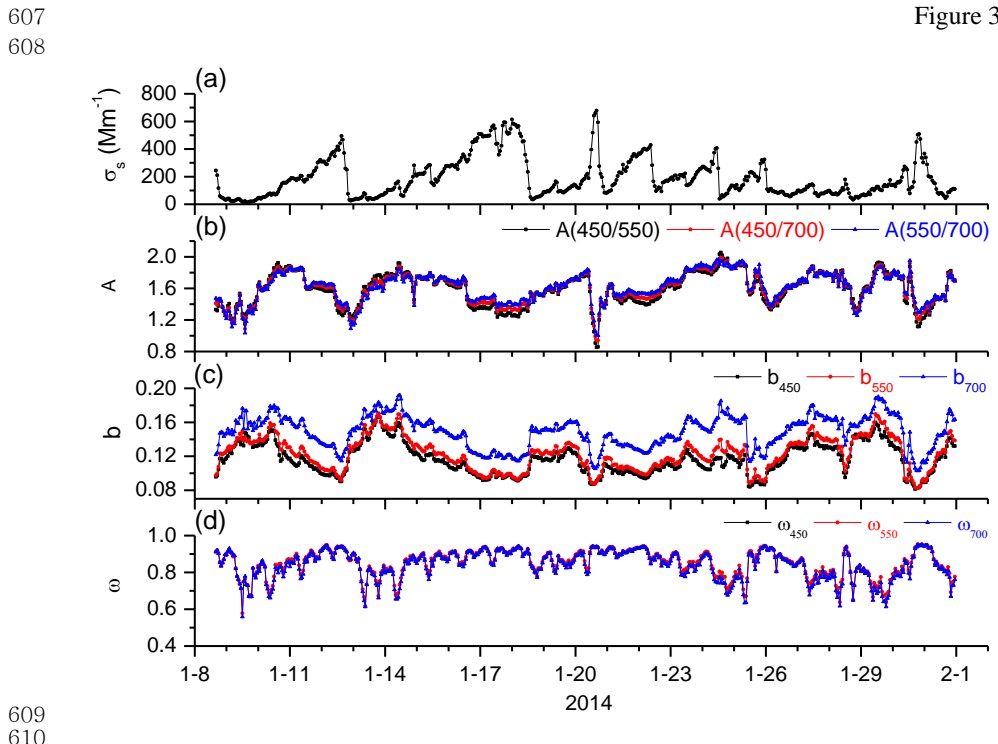






Figure 4

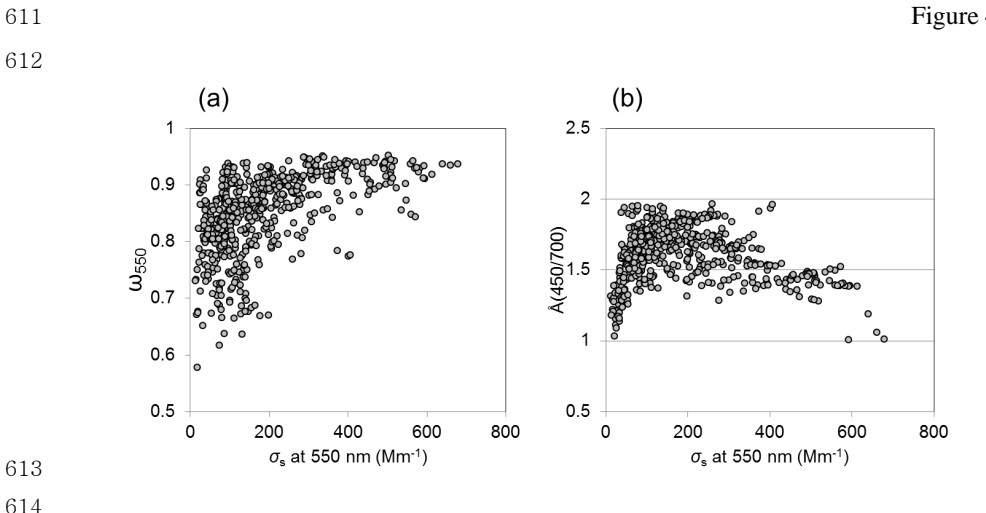





615                                                                                  Figure 5


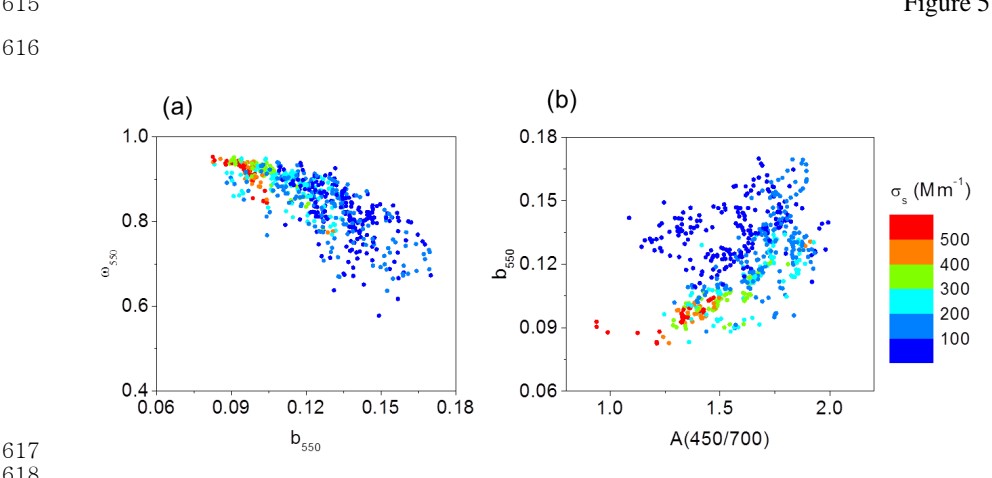








Figure 6

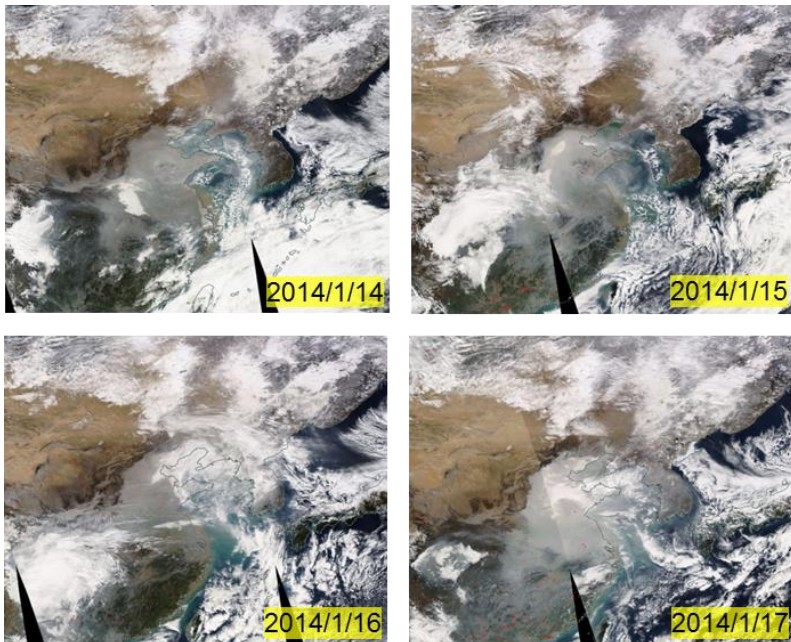








Figure 7


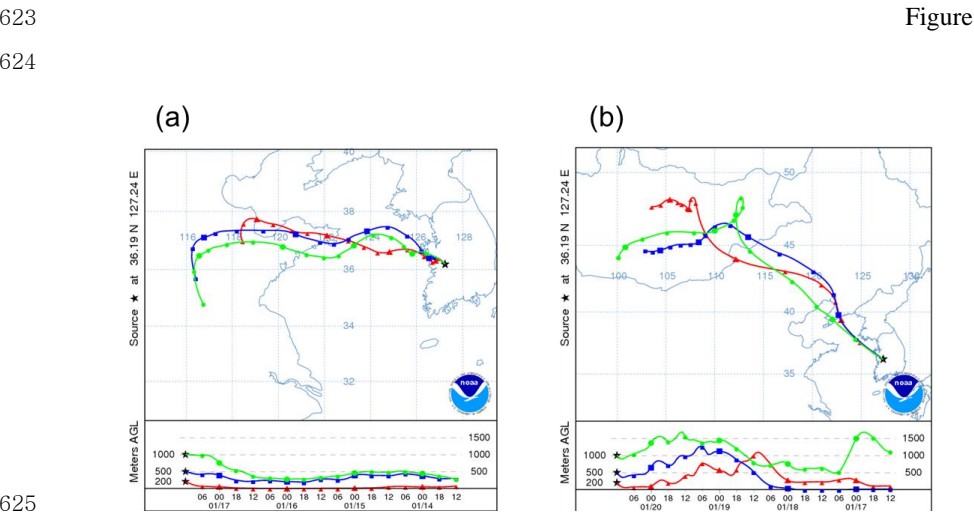





627                                                                                           Figure 8


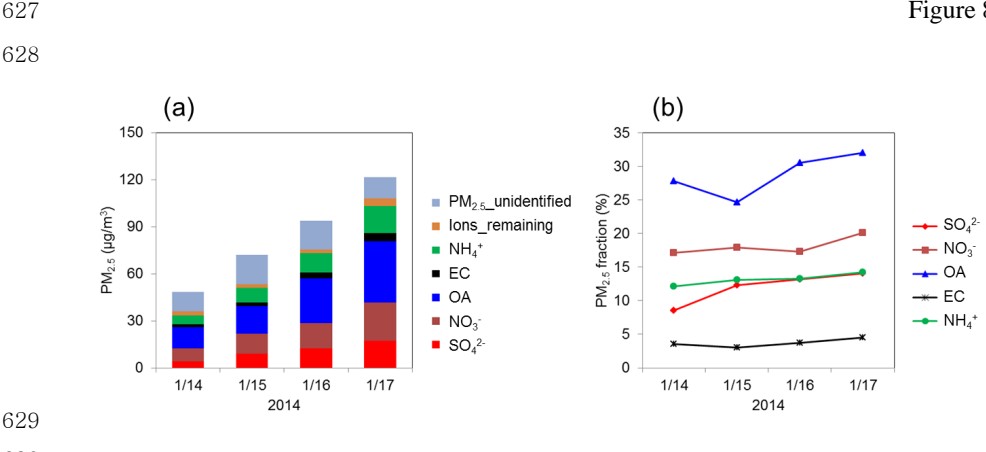







631                                                                                          Figure 9


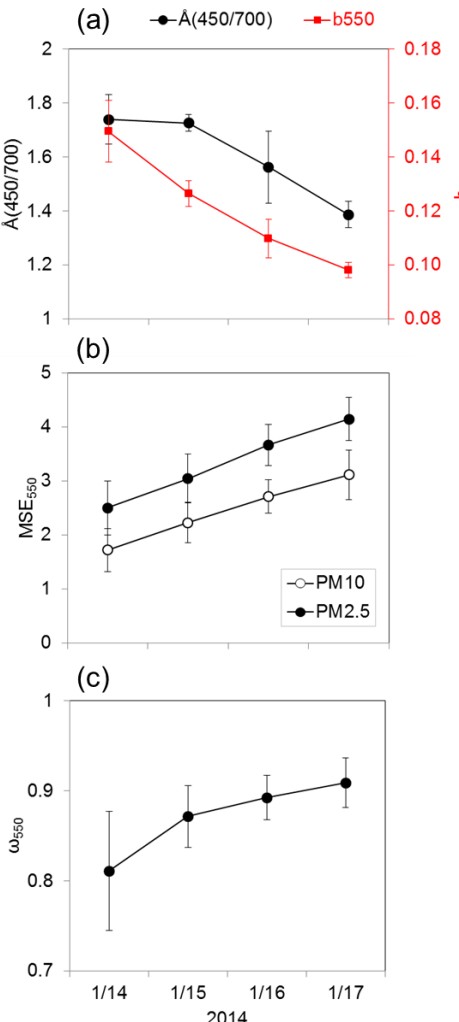




634                                                                                                    Figure 10


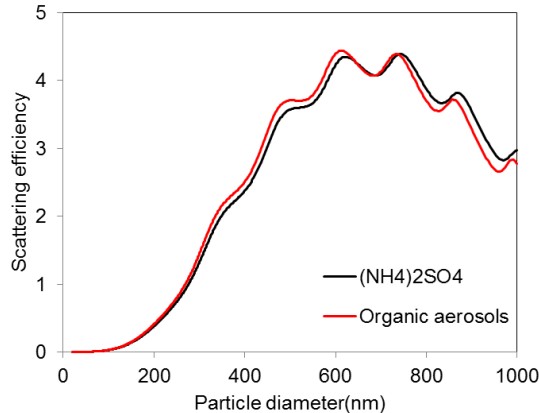







Figure 11

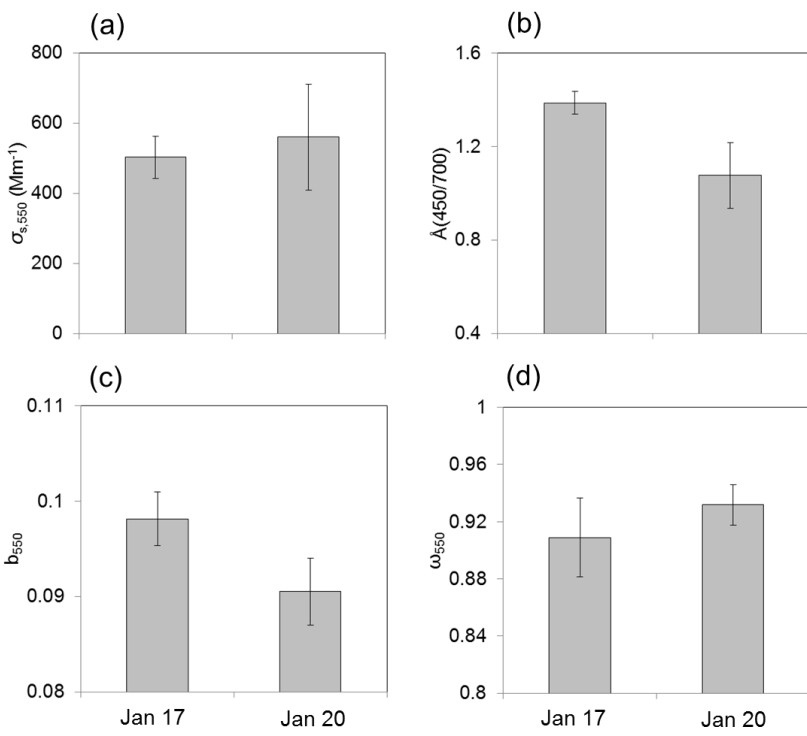

