# Peer review of "Ground based characterization of spectral optical properties of haze and Asian dust episodes under Asian continental outflow during winter 2014"

_Atmospheric Chemistry and Physics, 2016_

## Referee Comment (RC1) · Anonymous Referee #1 · 5 Dec 2016

General Comments Aerosol is very important to impact atmospheric cycle and climate system by direct and indirect effects, a hot issue of scientific researches internationally. Haze is a typical heavy pollution in East Asia, mainly caused by fine particles named as PM2.5. The growth of emissions and human activities with rapid urbanization is the most important reason for the increase of air pollution in this region. Also, the East Asia is an important source of dust, with large implications of regional environment and climate changes in downwind areas. The paper presents an intensive measurements of aerosol optical properties during one typical haze and Asian dust episodes in Korea, and analyze these optical properties and compare their difference between this two periods. In fact, high aerosol burden regions such as urban areas in Asia are still not well characterized in terms of particle amount. The topic of this paper is of common interest within the scientific community. Although the manuscript includes some important data, however, the quality is not sufficient in the current state to be directly published. The authors should take the suggestions made here into consideration for revision.

Specific suggestions 1. As for the title, it should be changed as "......of aerosol (spectral) optical properties......" . 2. In section 3.1, Figure 2 gives a temporal variation of wind, PM10, etc. the paper should address the time resolution of these values clearly, such as PM2.5, PM10, at hourly? Daily? 3. There are so many ground-based measurements of aerosol optical properties in East Asia. In fact this paper uses normal instruments, data and analyses as prior, however no clear new points. Although some available data, I suggest that the paper should add more deep analyses for aerosol mixing or aging due to transport using optical properties. 4. There is some room for revision in english.

---

## Referee Comment (RC2) · Anonymous Referee #2 · 21 Dec 2016

General Comment The manuscript discusses the characterisation of spectral optical properties of particles collected during haze and Asian haze episode in Daejeon, Republic of Korea during winter 2014. This study suggest PM2.5/PM10 mass ratios and Å(450/700) can be used as tracers to distinguish aged LRT haze and Asian dust under the Asian 39 continental outflow. Overall the study is very interesting unfortunately the information given in especially in the abstract and introduction still need to be improved. I suggest the authors to focus on the information haze and Asian dust episodes in their introduction. The period of study is too short to indicate the differences of these two episodes which only separated within few days. The comparison between haze

and Asian haze episode is not well discussed based on overall data including PM2.5 compositions.

Detail Comment 1. It's hard to understand information in the abstract clearly. I suggest the authors rewrite their abstract with clear problem statement, main objective, main methodology, main finding and conclusion from the research finding. 2. The introduction does not clearly present the problem statement of the research clearly. No explanation on the source haze and Asian dust and different characteristics of the particles from these two different sources from previous studies. 3. Line 61. Why only optical properties of LRT haze need to be studied. I suggest the authors write the need of the study such as in Line 51-53 and Line 61 -63, Line 70-72 in the last part of their introduction. 4. Line 85: Why this study only conducted during winter season? 5. Line 81-83: Any reason why the single scattering albedo was different when the air mass coming from different directions? 6. Line 85: Any particular reason on why this stud only conducted during winter? 7. The subtitle for section 2.1 is "Measurement site" but the authors explain about the online measurements of optical properties and manual sampling for PM2.5. I suggest the author to be more specific in their information under the sub-title. 8. Section 2.5: Is this section explain the determination of chemical composition of PM2.5? Please include the information in the title and main text for this sub-title. 9. On- month duration study with three episodes of haze can be considered short time for this kind of study. 10. How the authors define "haze condition" on 12, 17 and 20th January 2017? Haze usually relates with low visibility and high concentration of PM. Any cutting value for PM concentration? 11. Line 397: What the author mean by "stagnant atmospheric condition"? 12. Line 289: Why the authors only focus on second and third haze episodes? Any particular reason? 13. Line 298-299: What high concentration of K in PM indicates for the source of aerosols? Is K concentration based on measurement from PM2.5 compositions? Is there any mixture of haze and desert dust from this study? I asked this because the time period between these two episode are very close (only three days). 14. Line 305: Any explanation on why the compositions of PM2.5 were only measured until 17th of Jan 2014 (Figure 8). The composition

PM2.5 from third episode need to be measured for comparison. 15. Conclusion: What are the cutting points to indicate haze and Asian dust episode base on PM2.5/PM10 ratio and Å(450/700)? 16. Is there any influence of local source as one of the limitation of this study?

———————————————————

---

## Author Comment (AC1) · 3 Mar 2017

**Manuscript #: acp-2016-961**

**Title: Ground based characterization of spectral optical properties of haze and Asian dust episodes under Asian continental outflow during winter 2014**

Authors: Jinsang Jung et al.

**Responses to the reviewer's specific comments and questions;**

**Reviewer #1 (Comments):**

**General comments:**

Aerosol is very important to impact atmospheric cycle and climate system by direct and indirect effects, a hot issue of scientific researches internationally. Haze is a typical heavy pollution in East Asia, mainly caused by fine particles named as $PM_{2.5}$. The growth of emissions and human activities with rapid urbanization is the most important reason for the increase of air pollution in this region. Also, the East Asia is an important source of dust, with large implications of regional environment and climate changes in downwind areas. The paper presents an intensive measurement of aerosol optical properties during one typical haze and Asian dust episodes in Korea, and analyzes these optical properties and compares their difference between these two periods. In fact, high aerosol burden regions such as urban areas in Asia are still not well characterized in terms of particle amount. The topic of this paper is of common interest within the scientific community. Although the manuscript includes some important data, however, the quality is not sufficient in the current state to be directly published. The authors should take the suggestions made here into consideration for revision.

**Specific comments:**

*1. As for the title, it should be changed as ": : :: : :of aerosol (spectral) optical properties: : :: : :".*

**Response:** Thank you for the suggestion. The term "aerosol" has been added in the title of the MS. Please see line 1 in the revised MS.

*2. In section 3.1, Figure 2 gives a temporal variation of wind, PM10, etc. the paper should address the time resolution of these values clearly, such as PM2.5, PM10, at hourly? Daily?*

**Response:** Following sentence has been added in lines 152-153 in the revised MS.

"Hourly averaged mass concentrations of PM$_{2.5}$, Ca, OC, and EC were used in this study."

Following sentence has been added in lines 178-179 in the revised MS.
"Hourly averaged light scattering and absorption coefficients were used in this study."

Following sentence has been added in line 196 in the revised MS.
"Daily average water-soluble ions were used in this study."

The term "hourly average" has been added in Figs 2 and 3 captions.

*3. There are so many ground-based measurements of aerosol optical properties in East Asia. In fact this paper uses normal instruments, data and analyses as prior, however no clear new points. Although some available data, I suggest that the paper should add more deep analyses for aerosol mixing or aging due to transport using optical properties.*

**Response:** Thank you for the comments. We analyzed deeply about aerosol optical properties of the LRT haze and their aging under the stagnant atmospheric condition using aerosol chemical composition in chapter 3.3.2. Following paragraph has been added in lines 393-403 in the revised MS.

"Because the LRT haze from the Asian continent reached to the Korean Peninsula on 14 January as shown in Fig. 6, aerosol optical properties on 14 January can be used to investigate aerosol mixing state or aging during the atmospheric transport. When intensive optical properties of aerosols on 14 January was compared those obtained at the air mass source regions in China, no big difference between them was observed. For example, MSE$_{550}$ of PM$_{10}$ (1.73 ± 0.40 m$^2$ g$^{-1}$) on 14 January was similar to those (1.4 ± 0.89 m$^2$ g$^{-1}$) during relatively clean condition in Beijing, China but much lower than those (3.1 ± 0.9 m$^2$ g$^{-1}$) during relatively polluted condition (Jung et al., 2009a). $\omega_{550}$ (0.81 ± 0.07) on 14 January was also similar to those (~0.75) during relatively clean condition in Beijing. These results imply that aerosol aging is insignificant during the atmospheric transport from China to the Korean Peninsula in winter."

*4. There is some room for revision in English.*
**Response:** The revised MS has been proof-read by a native English speaker before submitting the revised MS.

---

## Author Comment (AC2) · 3 Mar 2017

**Manuscript #: acp-2016-961**

**Title: Ground based characterization of spectral optical properties of haze and Asian dust episodes under Asian continental outflow during winter 2014**

Authors: Jinsang Jung et al.

**Responses to the reviewer's specific comments and questions;**

**Reviewer #2 (Comments):**

**General comments:**

The manuscript discusses the characterization of spectral optical properties of particles collected during haze and Asian haze episode in Daejeon, Republic of Korea during winter 2014. This study suggest PM2.5/PM10 mass ratios and Å (450/700) can be used as tracers to distinguish aged LRT haze and Asian dust under the Asian continental outflow. Overall the study is very interesting unfortunately the information given in especially in the abstract and introduction still need to be improved. I suggest the authors to focus on the information haze and Asian dust episodes in their introduction. The period of study is too short to indicate the differences of these two episodes which only separated within few days. The comparison between haze and Asian haze episode is not well discussed based on overall data including PM2.5 compositions.

**Specific comments:**

1. It's hard to understand information in the abstract clearly. I suggest the authors rewrite their abstract with clear problem statement, main objective, main methodology, main finding and conclusion from the research finding.

**Response:** Following sentences added in the abstract in the revised MS.

"Long-range transported (LRT) haze can affect the regional radiation budget and the air quality in areas downwind of the Asian continental outflow. Because *in situ* observations of spectral aerosol optical properties of the LRT haze are rare, an intensive characterization of aerosol optical properties is needed. This study characterized the spectral optical properties of the LRT haze and Asian dust originating from the Asian continent. Integrated chemical and optical measurements of aerosol particles were carried out in a downwind area of the Asian continental outflow (Daejeon, Korea) during winter 2014." in lines 22-29. "These results imply that a change in particle size rather than chemical composition during the stagnant period is the dominant factor affecting the aerosol optical properties." in lines 40-42.

2. The introduction does not clearly present the problem statement of the research clearly. No explanation on the source haze and Asian dust and different characteristics of the particles from these two different sources from previous studies.

**Response:** Thank you for the comment. Following paragraphs has been added in lines 61-71 in the revised MS.

"With rapid economic growth and urbanization, megacities in China have experienced severe air pollution problems (Chan and Yao, 2008; Liu et al., 2013; Wang et al., 2014). In addition to anthropogenic pollutants, Asian dust originated from major deserts located in northern and western parts of China (e.g., Gobi desert and Taklimakan desert) influences the air quality of China (Bi et al., 2016; Li et al., 2016). Asian dust has highly light scattering property (single scattering albedo,  $\omega$  at 550 nm = 0.935) and low wavelength dependence of optical property (Å ngström exponent, Å at 440–870 nm = ~0.2) (Bi et al., 2016) whereas anthropogenic pollutants from mega-cities in China have relatively high light absorbing property ( $\omega$  at 532 nm = 0.82 (Guangzhou), 0.86 (Beijing), 0.83 (Shanghai)) and strong wavelength dependence (Å at 450–700 nm = 1.46 (Guangzhou), 1.42 (Beijing)) (Garland et al., 2008, 2009; Cheng et al., 2015)."

Four references have added in the reference section.

- Bi, J., Huang, J., Holben, B., and Zhang, G.: Comparison of key absorption and optical properties between pure and transported anthropogenic dust over East and Central Asia, Atmos. Chem. Phys., 16, 15501–15516, 2016
- Chan, C. K., and Yao, X.: Air pollution in mega cities in China, Atmos. Environ., 42, 1–42, 2008.
- Liu, X., et al.: Formation and evolution mechanism of regional haze: A case study in the megacity Beijing, China, Atmos. Chem. Phys., 13, 4501–4514, 2013.
- Wang, L., Wei, Z., Yang, J., Zhang, Y., Zhang, F., Su, J., Meng, C., and Zhang, Q.: The 2013 severe haze over southern Hebei, China: Model evaluation, source apportionment, and policy implications, Atmos. Chem. Phys., 14, 3151–3173, 2014.

3. Line 61. Why only optical properties of LRT haze need to be studied. I suggest the authors write the need of the study such as in Line 51-53 and Line 61 -63, Line 70-72 in the last part of their introduction.

Response: The terms "the chemical and" has been added in line 81 in the revised MS.

**4. Line 85: Why this study only conducted during winter season?**

**Response:** Because emissions of gaseous and particulate pollutants from the Asian continent increased during winter for space heating, we chose winter. Additionally, RH over the Korean Peninsula is low during winter which didn't cause measurement artifact of optical instruments.

**5. Line 81-83: Any reason why the single scattering albedo was different when the air mass coming from different directions?**

Response: Following sentence has been added in lines 103-104 in the revised MS.

"Garland et al. (2009) found that relatively low  $\omega$  for the air mass from the north was caused by the high emission of soot from combustion sources in Beijing."

**6. Line 85: Any particular reason on why this study only conducted during winter?**

**Response:** Because the emissions of gaseous and particulate pollutants from the Asian continent increased during winter for space heating, we chose winter. Additionally, RH over the Korean Peninsula is low during winter which didn't cause measurement artifact of optical instruments.

7. The subtitle for section 2.1 is "Measurement site" but the authors explain about the online measurements of optical properties and manual sampling for PM2.5. I suggest the author to be more specific in their information under the sub-title.

**Response:** The subtitle for section 2.1 has been changed to "General description of measurement" in line 115 in the revised MS.

8. Section 2.5: Is this section explain the determination of chemical composition of PM2.5? Please include the information in the title and main text for this sub-title.

**Response:** The section 2.5 contains satellite RGB images and air mass backward trajectories. The subtitle of section 2.4 has been modified as follows in line 181 in the revised MS. "Water-soluble ions analysis of PM2.5 filter samples"

9. One month duration study with three episodes of haze can be considered short time for this kind of study.

**Response:** Authors agree to the reviewer's opinion. However, because time resolution of optical properties of aerosol was hourly interval, we can investigate more detail about their temporal pattern. We also plan to investigate haze episode for long-term scale.

10. How the authors define "haze condition" on 12, 17 and 20th January 2017? Haze usually

relates with low visibility and high concentration of PM. Any cutting value for PM concentration?

**Response:** Following sentence has been added in lines 318-319 in the revised MS.

"This study defines haze episode when  $PM_{2.5}$  mass concentration is higher than 80 µg/m3 or  $PM_{10}$  mass concentration is 150 µg/m3."

11. Line 397: What the author mean by "stagnant atmospheric condition"?

**Response:** "stagnant atmospheric condition" means that air mass is stagnant due to low wind speed. Please see lines 261-263 in the revised MS.

The term "(< 1 m/sec)" has been added in line 263 in the revised MS.

12. Line 289: Why the authors only focus on second and third haze episodes? Any particular reason?

**Response:** This study tried to investigate temporal patterns of chemical and optical properties of LRT pollutants under the stagnant atmospheric condition. We also want to compare two haze episodes caused by urban pollutants from China and Asian dust. Because higher concentration of PM mass was observed during the second episode compared to the first episode, we choose the second episode.

13. Line 298-299: What high concentration of K in PM indicates for the source of aerosols? Is K concentration based on measurement from PM2.5 compositions? Is there any mixture of haze and desert dust from this study? I asked this because the time period between these two episodes are very close (only three days).

**Response:** Thank you so much for the comment. Authors did a mistake. The indicator of Asian dust should be not K (potassium) but Ca (calcium). Authors have changed K to Ca in the revised MS.

The term "K" was changed to "calcium (Ca)" in line 136 in the revised.

The minimum detection limit of Ca was changed to "0.32 ng m-3" in line 142 in the revised MS. The phrase starting "very high concentrations …" was modified as "Ca (maximum: 9.4  $\mu$ g m-3, average: 3.2 ± 3.4  $\mu$ g m-3)" in lines 329-330 in the revised MS.

The sentence starting "Elevated K concentrations ..." was revised as follows. Please see lines 409-410 in the revised MS.

"Elevated Ca concentrations were observed during the Asian dust episode, with an average of  $3.2 \pm 3.4 \ \mu g \ m^{-3}$ ."

In the Table 2: K concentrations during LRT haze and Asian dust episodes have been changed to Ca concentrations. Please see modified Table 2 in the revised MS.

14. Line 305: Any explanation on why the compositions of PM2.5 were only measured until 17th of Jan 2014 (Figure 8). The composition PM2.5 from third episode need to be measured for comparison.

**Response:** Thank you for the comment. Chemical composition of PM2.5 was measured during the entire measurement period. Aerosol sampling was conducted every 24 hr whereas Asian dust occurred only for 6 hr on 20 January. So, daily averaged chemical composition can't represent Asian dust. Thus, we didn't compare chemical compositions of Asian dust and LRT haze in this study.

15. Conclusion: What are the cutting points to indicate haze and Asian dust episode base on *PM2.5/PM10* ratio and Å(450/700)?

**Response:** Following sentence has been added in lines 432-434 in the revised MS.

"This study suggests that  $PM_{2.5}/PM_{10}$  mass ratio and Å(450/700) of <0.6 and <1.0, respectively, can be used as the cutting points to indicate Asian dust mixed with haze."

**16. Is there any influence of local source as one of the limitation of this study?**

**Response:** Following paragraph has been added in lines 461-467 in the revised MS.

"The results of this study imply that severe haze episodes over the Korean Peninsula are mainly caused by long-range transported pollutants from the Asian continent. These severe haze episodes can be enhanced under the stagnant atmospheric condition. It is postulated that emissions from local sources can also contribute to severe haze episodes under the stagnant atmospheric condition. Thus, the contribution of local sources to severe haze episodes needs to be classified and quantified in a future study to better understand the optical property of aerosols."

---

## Author Response (AR2)

**Manuscript #: acp-2016-961**

**Title: Ground based characterization of spectral optical properties of haze and Asian dust episodes under Asian continental outflow during winter 2014**

Authors: Jinsang Jung et al.

**Responses to the reviewer's specific comments and questions;**

**Reviewer #1 (Comments):**

**General comments:**

The manuscript has been been improved a lot. It can be accepted after minor revisions:

**Specific comments:**

1. The authors need to clarify in the last part of introduction on why PM10 and PM2.5 need to be measured for the haze optical measurement. They also needs to mention why this experiment was focused in winter which I think due to wind direction from mainland China.

**Response:** The terms "Size-segregated mass," has been added in lines 108-109 in the revised MS.

Following sentence has been added in lines 107-108 in the revised MS.

"Because fossil fuel consumption increases during winter for space heating and northwesterly winds are dominant during winter, this study focused on winter."

2. Line 153: Remove "Ca" because it has been mentioned in the previous paragrapgh

**Response:** The term "Ca" in line 153 in the original MS was removed.

3. Line 198: Include full name of "RGB" when it was first mentioned in the main text

**Response:** The term "(Red, Green, Blue)" has been added in line 200 in the revised MS.

4. On what basis were PM2.5 with concentration of >80 ug/m3 and PM10 = 150 used to determine haze episode? The value of PM2,5 far exceeds the WHO/USEPA standard. PM10 should be written as "PM10 concentration is higher than 150 ug/m3".

**Response:** Following sentence has been added in lines 321-323 in the revised MS.

"These threshold values of the haze episode correspond to visibility of 8 km estimated by Jung et al. (2009b)."

The term "higher than" has been added in line 321 in the revised MS.